# Ran GTPase and Its Importance in Cellular Signaling and Malignant Phenotype

**DOI:** 10.3390/ijms24043065

**Published:** 2023-02-04

**Authors:** Mohamed El-Tanani, Hamdi Nsairat, Vijay Mishra, Yachana Mishra, Alaa A. A. Aljabali, Ángel Serrano-Aroca, Murtaza M. Tambuwala

**Affiliations:** 1Pharmacological and Diagnostic Research Centre, Faculty of Pharmacy, Al-Ahliyya Amman University, Amman 19328, Jordan; 2School of Pharmaceutical Sciences, Lovely Professional University, Phagwara 144411, India; 3Department of Zoology, School of Bioengineering and Biosciences, Lovely Professional University, Phagwara 144411, India; 4Department of Pharmaceutics & Pharmaceutical Technology, Yarmouk University, Irbid 21163, Jordan; 5Biomaterials and Bioengineering Laboratory, Centro de Investigación Traslacional San Alberto Magno, Universidad Católica de Valencia San Vicente Mártir, c/Guillem de Castro 94, 46001 Valencia, Spain; 6Lincoln Medical School, University of Lincoln, Brayford Pool, Lincoln LN6 7TS, UK

**Keywords:** Ran GTPase, mitosis, metastasis, osteopontin, aneuploidy

## Abstract

Ran is a member of the Ras superfamily of proteins, which primarily regulates nucleocytoplasmic trafficking and mediates mitosis by regulating spindle formation and nuclear envelope (NE) reassembly. Therefore, Ran is an integral cell fate determinant. It has been demonstrated that aberrant Ran expression in cancer is a result of upstream dysregulation of the expression of various factors, such as osteopontin (OPN), and aberrant activation of various signaling pathways, including the extracellular-regulated kinase/mitogen-activated protein kinase (ERK/MEK) and phosphatidylinositol 3-kinase/Protein kinase B (PI3K/Akt) pathways. In vitro, Ran overexpression has severe effects on the cell phenotype, altering proliferation, adhesion, colony density, and invasion. Therefore, Ran overexpression has been identified in numerous types of cancer and has been shown to correlate with tumor grade and the degree of metastasis present in various cancers. The increased malignancy and invasiveness have been attributed to multiple mechanisms. Increased dependence on Ran for spindle formation and mitosis is a consequence of the upregulation of these pathways and the ensuing overexpression of Ran, which increases cellular dependence on Ran for survival. This increases the sensitivity of cells to changes in Ran concentration, with ablation being associated with aneuploidy, cell cycle arrest, and ultimately, cell death. It has also been demonstrated that Ran dysregulation influences nucleocytoplasmic transport, leading to transcription factor misallocation. Consequently, patients with tumors that overexpress Ran have been shown to have a higher malignancy rate and a shorter survival time compared to their counterparts.

## 1. Introduction

### 1.1. The Roles of Ran within the Cell

Ran is a member of the Ras superfamily of proteins. Ran exists in two states, Ran GTP and Ran GDP, forming the Ran cycle. The balance between these two states is maintained by the action of the guanine nucleotide exchange factor of regulator of chromosome condensation 1 (RCC1), converting GDP to GTP, and the activation of Ran’s intrinsic GTPase activity through association with Ran guanine activating proteins (Ran GAP) and Ran binding proteins, leading to the hydrolysis of GTP [1,2,3]. Through the action of this cycle, Ran acts as a regulator of normal cellular function, mediating several important processes [4].

### 1.2. The Role of Ran in Nucleocytoplasmic Transport and Cell Cycle Progression

The expression of Ran has also been shown to play a vital role in the regulation of mitosis and nucleocytoplasmic transport in tumor cells, further implicating it as a mediator of tumor cell survival [5,6,7,8]. Ran plays an important role in regulating the directionality of nucleocytoplasmic transport. The difference in concentrations of Ran GTP and Ran GDP in the nucleus facilitates the import and export of vital molecules, presenting nuclear localization signals (NLSs) and nuclear export signals (NESs), including proteins and RNA [1,9,10,11]. Thus, Ran is essential for signaling from the nucleus to the cytoplasm and vice versa [12,13,14]. Ran also regulates microtubule formation through the release of several effector molecules in the proximity of the chromosomes [15,16,17,18]. This is linked to its role in transport and influences the progression of the cell cycle and the maintenance of regular cell division and cell integrity [19,20,21,22]. Finally, Ran facilitates NE reassembly after cell division, further solidifying Ran’s essentiality in normal cell division [20,23,24].

### 1.3. The Role of Ran in Cancer Progression

Furthermore, Ran has been identified as a downstream effector in multiple signaling pathways that are frequently disrupted in various types of cancer. The disruption of these pathways plays an important role in increasing tumor cell dependence on Ran, and this has a significant impact on the regulation of tumor cell survival [5,6,8,25]. Ran has been implicated in multiple human cancer cell lines [7,26,27]. It is thought to play a contributory role in tumor formation in various types of human cancer. The implantation of Ran overexpressing cells has been shown to be tumorigenic in vivo [27]. Furthermore, the overexpression of Ran has been observed in numerous human cancer cell lines [7,26,27,28]. Moreover, Ran is an effector molecule of several growth factors, which have been shown to play a significant role in some cancer types [29,30]. Any changes in the expression of these genes have a downstream effect on Ran expression, therefore suggesting a pathway for tumorigenesis [7,26,31].

These alterations may play an important role in cancer, resulting in the formation of tumors and a rise in malignancy in correlation to Ran’s overexpression [32]. The aberrant expression of Ran causes the dysregulation of several processes within the cells, leading to altered cellular function. Cells expressing high Ran levels display several fundamental changes: An increased ability to proliferate, including the loss of contact inhibition, which leads to increased tissue density, and the ability to grow in an anchorage-independent manner [3,26,27,29,33,34].

## 2. Ran Regulates Nucleocytoplasmic Transport

The difference in Ran GTP concentration between the nucleus and cytoplasm determines transport direction. GTP to GDP conversion controls Ran’s activation [35,36]. Ran’s GTPase hydrolyzes GTP into GDP [35,37,38]. RanGAP1, RanBP1, and RanBP2 increase the hydrolysis rate of Ran-GTPase [1,39], and RCC1 regulates the GDP-GTP exchange [40]. During the interphase, RanGAP1, RanBP1, and RanBP2 are in the cytoplasm, while RCC1 is chromatin-associated and anchored in the nucleus [2,41,42]. Nuclear transport requires Ran [43]. Ran-GTP-binding proteins act as nuclear import and export receptors. Import receptors bind their cargo in the cytosol, where Ran-GTP concentrations are low, and dissociate in the nucleus. Export receptors bind cargo in Ran-GTP complexes in the nucleus and dissociate in the cytoplasm following GTP hydrolysis. The import of proteins with traditional NLS is mediated by a heterodimeric receptor consisting of importin and importin (Figure 1). Specific localization of these enzymes increases Ran-GTP in the nucleus and Ran-GDP in the cytoplasm [44,45,46]. This gradient is essential to Ran’s activity, allowing the synthesis and disassembly of cargo protein complexes [47]. Ran-GTP binds CRM1 in the nucleus during export. Conformational alteration allows the binding of nuclear export signal-containing proteins. Nuclear pore complexes transfer complexes to the cytoplasm [46]. Ran-GTP is hydrolyzed by RanGAP1, RanBP1, and RanBP2 [13,48,49] and dissociates the complex. Cargo proteins with a nuclear localization signal trigger import. In its presence, importin and the adaptor protein importin form a heterodimeric complex that binds the NLS. The complex allows NPC trafficking into the nucleus and Ran GTP enhances complex dissociation in the nucleus [14,50]. After transport, Ran GTP is hydrolyzed, dissociating the Ran GTP–importin complex. Importin is regenerated with the cellular apoptosis susceptibility gene product (CAS) [51]. Ran GDP import and conversion to GTP are regulated by a different route via NTF2. These processes maintain the Ran gradient and binding/releasing partner distribution [52].

### Molecules Transported

Ran-directed formation and dissociation of exportin complexes are key to exporting RNAs, ribonucleoproteins (RNPs), and proteins from the nucleus [53,54]. The transport of RNA involves the shuttling of several forms of RNA: tRNA is required for ribosomal translation of mRNA, microRNAs (miRNAs) participate in gene regulation, initiating post-transcriptional gene silencing by binding to the target mRNA, Small nuclear RNAs (snRNAs) are involved in the splicing of pre-mRNAs through the removal of introns, and rRNAs are involved in the formation of ribosomes, which provide the mechanism for protein translation [55,56]. Ran-dependent import pathways transfer proteins into the nucleus, such as nucleoplasmin. Other import targets include U snRNPs, which splice during transcription [56,57]. Various transcription factors are transported into the nucleus in a Ran-dependent way, and several of these factors affect cellular proliferation and differentiation. Translocation of STAT-3 requires importin-β1 and Ran. STAT-3 stimulates cellular differentiation, proliferation, and tumor cell invasion [58]. Therefore, the disturbance in Ran expression is linked to phenotype transformation, which may lead to the development of cancer and metastasis.

## 3. Ran Regulates Spindle Formation

It has been shown that the Ran GTP concentration in the nucleus controls mitotic progression. The cell’s microtubule system is modified during mitosis to generate the spindle apparatus [59]. Ran GTP has been demonstrated to induce the production of microtubule asters surrounding centrosomes in vitro, whereas RanT24N, which is predominantly coupled to GDP, lacks this capacity [60]. In addition, mutations in RCC1 and RanBP1 have been reported to cause cell cycle arrest due to microtubule misalignment [17,59]. Normal conditions result in the formation of a bipolar spindle in cells with duplicate centrosomes, which is characterized by the nucleation of numerous longer microtubules by the centrosomes, which have a strong affinity for chromosomes and are therefore oriented towards them until only a bipolar spindle remains [61,62]. Slight centrosomal aster development was detected after the inclusion of the Ran mutant [63]. In addition, these microtubules did not interact with the chromosomes, and as a result, numerous centrosomes were in groups with no microtubule binding. This evidence demonstrates that inhibiting Ran GTP production has a negative effect on microtubule assembly surrounding chromosomes [64,65,66].

It is assumed that chromatin plays a role in spindle formation [65,67]. Ran GTP has a concentration gradient surrounding the chromosome, which is mediated by RCC1’s association with chromatin (Figure 2). Local Ran GTP concentrations have been demonstrated to increase microtubule nucleation and stability [68,69]. The antagonistic connection between anchored RCC1 and unbound RanBP1 and Ran GAP establishes the concentration gradient, resulting in a localized high Ran GTP concentration surrounding the chromosomes. The inability of chromatin to nucleate microtubules was eliminated by inhibiting the production of Ran GTP [70,71]. This mechanism is believed to rely on the Ran GTP-dependent dissociation of components from importin, such as targeting protein X2 (TPX2), which has been demonstrated to increase the nucleation of microtubules near chromosomes [18,72,73].

### 3.1. Mechanism of Spindle Regulation by Ran

Importin α and β complexes interact with Ran GTP as the fundamental mechanism underpinning Ran’s regulatory function. This association neutralizes the inhibitory impact of importin and, therefore, releases spindle assembly components from the complex [74,75]. It has been demonstrated that the local separation of these components from importin complexes promotes microtubule nucleation and stability in close proximity to chromosomes [76,77]. Importin β-Ran GTP and the nuclear mitotic apparatus protein (NuMA) interaction establishes a link between Ran signaling and NuMA function [66,78,79] and demonstrates that importins can sequester NuMA, preventing it from interacting with microtubules, except in regions of high Ran GTP concentrations [80]. It has been demonstrated that TPX2, when added in excess relative to importin concentration, spontaneously initiates aster formation. M phase spindle development requires TPX2 for the stimulation of Ran GTP and chromatin-mediated spindle formation. TPX2 is inactive because it is linked to importin-β in the absence of Ran GTP. When Ran GTP binds to importin-, it makes it easier to separate the factor TPX2 from importin-. As previously discussed, this results in the formation of microtubules surrounding chromosomes [81,82,83,84]. TPX2 is also required for the production of K-fibers [85]. In addition, Ran GTP promotes the connection between TPX2 and Eg2, activating Eg2 and controlling spindle assembly [73,86]. Numerous other spindle assembly factors (SAFs) have been discovered; HURP is required for the stability of K-fibers and binds bundles. During the interphase, importin- has been shown to transport HURP from the cytosol to the nucleus, and its concentration is controlled by its fast export via CRM1. The dissociation of HURP from importin is reliant on Ran and requires interaction with GTP-bound Ran. Consequently, the interaction of HURP with the spindle is dependent on the concentration of Ran GTP [87]. A Ran-dependent complex including HURP and TPX2 is also believed to be involved in microtubule formation; this complex also contains the microtubule-associated factor XMAP215 and the motor protein Eg5, which has been demonstrated to be essential for bipolar spindle assembly (Figure 3). XMAP215 is involved in the nucleation of microtubules [81,88]. Ran plays a key regulatory function in mitotic spindle formation and control through the dissociation of SAFs from importin-β and the creation of complexes [85].

### 3.2. Ran Regulates Nuclear Envelope Reassembly

Mitotic division also requires regulated reassembly of the nuclear membrane, which separates the nucleus and cytoplasm. Ran GTPase regulates this process. NE construction requires both RCC1-generated RanGTP and Ran’s GTPase activity [24]. Ran depletion in C. elegans induces NE defects. These data implicate the Ran cycle as a modulator of NE reassembly, either by vesicle fusion or chromatin modification. Ran induces NPC-containing NE assembly in sepherose beads without chromatin [89].

## 4. The Overexpression of Ran Alters Cellular Growth and Proliferation and Is Present in Cancer

Upon transfection of fibroblasts with the constitutively active Ran mutant F35A [90], it was found that Ran overexpression has a significant effect on cell proliferation. The transfected cells exhibited a greater capacity for proliferation, a lack of contact inhibition, and a higher cell density than the wild-type cells. Additionally, cells were able to develop in an anchorage-free environment. Additionally, animals injected subcutaneously with cells transfected with constitutively active Ran developed tumors, indicating its role in neoplastic transformation [27]. The Ran cycle is essential for regular cell activity; therefore, when Ran is expressed abnormally, as it is in certain tumor cells, aberrant cell growth and cell division are stimulated. In contrast to the modest expression reported in normal tissue, tumor cells express a significantly higher quantity of Ran GTP. In addition, Ran is present ubiquitously in aberrant tissue [21,22]. The overexpression of Ran was elevated at both the mRNA and protein levels in gastric adenocarcinomas. In contrast, normal cells within the tumor tissue did not exhibit higher levels of Ran [91]. In mouse tissue, it was discovered that lymphoma cells entering the liver had elevated levels of Ran, whereas surrounding hepatocytes had normal levels. In addition, Ran expression was found to be significantly higher in a variety of human malignancy tissues compared to normal tissue, such as in colon, breast, lung, and renal cell adenocarcinomas [21,22].

### 4.1. Ran Overexpression

The overexpression of Ran is associated with increased malignancy and invasiveness. This correlation was determined by comparing the levels of gene expression in non-invasive and invasive cell lines. Ran expression was elevated in invasive cell lines in comparison to non-invasive cell lines. The increase in Ran expression between cell lines corresponded to the difference in OPN expression [31]. This shows that Ran expression is associated with osteopontin expression. By transfecting an OPN expression vector into benign, non-invasive breast Rama 37 cells, the mRNA and protein levels of OPN and RAN were shown to be increased. Transfection with a Ran expression vector enhanced Ran expression but had no effect on OPN levels. Moreover, the transfection of OPN-overexpressing cells using OPN antisense cDNA decreased the levels of both OPN and Ran proteins, whereas small interfering RNA (siRNA)-RAN exclusively reduced Ran protein levels. These results indicated that Ran is an effector downstream of OPN [26,27]. This showed that the overexpression of osteopontin and the subsequent overexpression of Ran plays a crucial role in neoplastic transformation. In vitro, the overexpression of Ran increases cell adhesion, colony formation rate, and invasive potential [30,92]. Moreover, the implanting of Ran and OPN-expressing cells into rats resulted in the development of tumors with metastases [30]. Cells transfected with siRNA-Ran and cells expressing osteopontin and Ran had significantly different rates of tumor formation compared to untransfected cells. In addition, the silencing of Ran resulted in lower levels of cell invasion in vitro and in vivo compared to cells with a constitutive Ran expression vector [26].

#### 4.1.1. Ran Overexpression and Malignancy in Human Cancers

Increased osteopontin expression has been recognized as a critical factor in malignant transformation and cancer. In conjunction with the overexpression of Ran in osteopontin-transformed Rama37 cells, Ran has been identified as an osteopontin effector. Consequently, Ran overexpression is related to enhanced mammary epithelial invasion and metastases [26,93,94]. In addition, Ran overexpression has been seen in a number of different cancer types [21,22]. Overexpression has also been identified in acinar clear cell carcinoma, in which tumor cells express a considerably higher amount than the normal kidney tissue surrounding them. Immunohistochemistry staining demonstrated that the level of Ran in the nuclei of tumor cells was significantly greater than that of normal cells. The change in expression was similarly correlated to the tumor grade and, subsequently, the worsening of the prognosis, with sarcomatoid features exhibiting the highest levels of Ran expression. The presence of sarcomatoid markers indicates a poorer prognosis. Patients with metastases had higher levels of Ran than those without [95]. In general, across all grades of tumors, Ran was expressed at a higher level in metastatic malignancies [96,97]. Ran expression is also associated with increased cancer risk in epithelial ovarian tumors. This is because Ran and RanBP1 are more frequently overexpressed in invasive tumors than in those with low cancer risk [3,98,99,100].

#### 4.1.2. Ran Expression and Survival Time

The link between Ran and the malignant phenotype can be determined by analyzing variations in prognosis in cancer patients [32]. A survival curve was generated for patients with breast, lung, melanoma, and renal clear cell carcinoma. Based on the average expression level, these patients were assigned either a high or low expression level. Through this research, increased Ran expression was found to be associated with decreased overall survival. In patients without metastases undergoing nephrectomy or nephron-sparing surgery, higher expression was likewise associated with shorter disease-free survival [96]. Ran expression has also been connected to a decreased median survival time in patients with breast cancer [101]. Higher Ran expression in the nuclei and cytoplasm of malignant cells within primary tumors was related to a shorter survival time [102]. This was also demonstrated to be independent of other parameters, including tumor size, grade, and lymph node involvement. Ran expression in lung cancer resembled these findings; increased Ran expression was associated with a shorter survival time. In tumors with mutations or overexpression of proteins in several signaling pathways, a high level of Ran expression was associated with a decreased survival time. This includes the PIK3CA mutation found in breast cancer patients, which activates the PI3K/Akt/mTORC1 pathway [103,104]. In lung cancer, increased Ran expression drastically decreased the median survival time of individuals who overexpressed the mesenchymal–epithelial transition factor (c-Met) [105,106]. Similarly, increased Ran expression significantly decreased survival time in patients’ tumors that expressed OPN. In addition, patients with an elevated amount of Ran-coding mRNA in both breast and lung cancer had a worse prognosis than those with a low expression level. These outcomes were also reported in colorectal cancer patients with PI3K and K-Ras mutations; increased Ran expression was associated with a shorter survival time [6,107,108]. Moreover, Ran is overexpressed in numerous human cancers, including those of the stomach [91,109], lung [6,8], head and neck [110], pancreas [111], ovary [112], malignant melanoma [5], colorectal [113], and kidney [96], but not in non-tumor tissue [5].

## 5. Mechanism of Altered Expression—Ran Is a Downstream Effector of the PI3K/Akt and MEK/ERK Pathways

Various tumor cell types are dependent on oncogenic pathways for growth, proliferation, and prolonged survival. In cancer, the PI3K/Akt/mTORC1 and Ras/MEK/ERK pathways are both frequently hyperactivated [6,114,115]. The mechanisms that lead to alterations in Ran expression are not well understood. However, Ran has been shown to be an effector of multiple signaling pathways implicated in cancer. Ran is a downstream effector of Akt; hydrogen peroxide, which possesses the ability to induce mitosis in cells, has been shown to induce the phosphorylation of Akt and subsequently increase Ran expression [116]. Furthermore, the inhibition of PI3K prevented the induction of mitosis and prevented the phosphorylation of Akt and the subsequent downstream effects on Ran expression, demonstrating that PI3K is involved upstream [117]. When breast cancer cells were treated with separate inhibitors for both the PI3K and MEK pathways, silencing of Ran resulted in a greater degree of apoptosis than under normal conditions, suggesting that the abnormal activation of these pathways leads to tumor cell dependence on Ran [6,118,119].

### 5.1. Aberrant Control of Pathways and Tumor Cell Dependence on Ran

Ran silencing in cells expressing the K-Ras mutant led to a significant increase in apoptosis in comparison to those expressing the wild-type gene. In addition, the inhibition of PI3K or MEK dramatically reduced the apoptosis of these cells [6,120,121]. Higher levels of phosphorylated Akt and c-Met were found in cells transfected with Ran expression vectors [26]. c-Met has been demonstrated to activate PI3K, hence influencing the activation of Akt downstream targets [122], and has also been shown to depend on the Erk pathway [123]. Moreover, c-Met has been linked to lung, breast, and colon cancer, and others [124]. Several mechanisms for this effect have been found, including overexpression by gene amplification and the secretion of growth factors and activation via persistent activation of kinases [125]. As demonstrated in breast cell carcinoma, the phosphorylation of c-Met is related to the activation of many downstream signaling pathways, resulting in increased tumor cell motility and invasion potential [126,127]. Ran silencing was performed in two cell lines to study the role of c-Met in tumor cell survival: HCC827 cells, which have high levels of c-Met phosphorylation, and HCC87 GR5 (GR5) cells, which overexpress total and phosphorylated c-Met [128,129]. Ran silencing resulted in increased levels of apoptosis in GR5 cells, indicating that cells with amplified c-Met are more susceptible to Ran-silencing-induced apoptosis and are consequently more dependent on Ran for survival [130]. Ran has a tendency to induce apoptosis to a greater extent in abnormal cells, resulting in the abnormal activation of these pathways [118,131]. This shows that Ran is a downstream effector of these pathways, and that the hyperactivity of these pathways induces the survival dependence of tumor cells on Ran.

### 5.2. Signaling Pathways Phosphorylation of Ran Binding Proteins and Control of Ran Expression

In addition to the direct effect of signaling pathways on Ran expression, aberrant signaling pathway regulation was shown to indirectly influence the Ran gradient through phosphorylation and, therefore, the regulation of the activity of Ran binding protein 3 (RanBP3). Phosphorylation occurs through the activity of RSK and Akt, which are downstream from Ras/ERK and PI3K, respectively. RSK has been shown to possess the ability to phosphorylate RanBP3 [132,133]. Additionally, the knockdown of RSK1 and 2 blocked RanBP3 phosphorylation. Akt can also phosphorylate RanBP3. Similar to RSK, the addition of a PI3K inhibitor prevented the phosphorylation of RanBP3 [132]. Results suggest that phosphorylation may increase RanBP3′s binding affinity for Ran; increased interaction between the two was observed upon serum addition. This interaction was also inhibited by the addition of PI3K and MEK/ERK inhibitors. RanBP3 phosphorylation alters the nucleocytoplasmic balance of Ran; this was demonstrated through the knockdown of Ran BP3. In control cells, Ran GTP was distributed primarily within the nucleus, whereas in knockdown cells, a significant amount of Ran GTP was located in the cytoplasm. This is due to the inhibitory effect of phosphorylated RanBP3 on RCC1 and has consequences for cellular function [132].

### 5.3. Altered Ran Expression and Regulation of Cellular Function and Nucleocytoplasmic Transport

The phosphorylation of RanBP3 and subsequent changes in relative Ran concentrations had a profound effect on the efficiency of nucleocytoplasmic transport, as demonstrated through the analysis of the transport of ribosomal protein L12. Knockdown cells expressing the RanBP3 mutant S58A demonstrated reduced transport ability when compared with wild-type RanBP3. Additionally, knockdown cells expressing the mutant displayed reduced cell proliferation relative to the control cells [132]. Ran’s role in nucleocytoplasmic transport was identified by silencing it, as this had a profound effect on the distribution of transcription factors. Ran appears to have a significant influence on nucleocytoplasmic transport within cancer cells, altering the relative concentration of certain transcription factors within the nucleus and cytoplasm. This dysregulation appears to be dependent on active PI3K/Akt and MEK/ERK pathways in cancer cells. Ran silencing increased the nuclear localization of p53, p27, and C-jun while decreasing the nuclear localization of β-catenin. In breast cancer cells, Ran silencing decreased -catenin and NFκB nuclear localization while increasing p53 and p27 localization [134]. The PI3K inhibitor PI103 and the Akt pathway inhibitor rapamycin partially reversed the altered localization of β-catenin, p53, and p27 in lung cancer cells. The altered localization of β-catenin and p27 in breast cancer cells was reduced by the addition of PD184352, an MEK1 inhibitor, and rapamycin [135]. These data suggest that Ran expression can alter nucleocytoplasmic transport, thus influencing transcription factor distribution in cancer cells. This appears to be dependent on the activation of the MEK/EKT and PIK3/AKT pathways [6].

### 5.4. Effect of Ran Expression on Spindle Formation and Tumor Cell Survival

Ran expression appears to operate as a mitotic mediator in tumor cells, a crucial factor in the survival of cancer cells. Due to the loss of Ran-regulated microtubule production, it has been shown that Ran inhibition leads to abnormal mitosis and eventual cell death in breast cancer cells [29]. Abnormal mitosis was linked to the flattening of mitotic spindles, the depletion of microtubules, the irregular segregation of chromosomes, and the lack of TPX2 release. The cells died because of the production of hypodiploid DNA. Comparatively, the silencing of Ran in several types of normal cells exhibited no substantial effect on cell-cycle progression or cell viability [21,22]. It has also been demonstrated that Ran-dependent mitosis is necessary for the survival of K-Ras-activated cancer cells. Using siRNA to silence Ran and TPX2 in these cells resulted in decreased survival relative to cells in which the K-Ras gene was disrupted [136]. Ran and TPX2 knockdown led to cell cycle arrest and subsequent cell death [120]. It has also been demonstrated that a Ran-Survivin complex pathway is favored by tumor cells. The disruption of the formation of survivin-Ran complexes impeded the delivery of the effector TPX2 to microtubules in tumor cells, resulting in uneven mitotic spindle formation and perturbed chromosomal separation. This had no effect on normal cells, suggesting that tumor cells use this pathway over the traditional Ran pathway to promote their survival [21,22]. In epithelial ovarian cancer (EOC), the loss of Ran from EOC cells leads to the activation of apoptosis [112], which provides more evidence for the dependence of tumor growth and progression on Ran.

Moreover, it has been demonstrated that Ran is a new effector of multiple genes involved in spindle control. One such gene is the tumor suppressor gene RASSF1A. RASSF1A causes phosphorylation of RCC1 in the normal phenotype, resulting in the buildup of RanGTP [137,138]. This promotes the hyperstability of microtubules. The promoter region of this gene becomes hypermethylated in various forms of cancer, resulting in an increased incidence of tumor growth in mice. The reduction of RASSF1A causes misallocation of RCC1, which causes a buildup of Ran GTP around the mitotic spindle and at spindle poles [138,139]. This induces improper spindle assembly, resulting in aneuploidy and eventual tumor growth. This suggests that Ran plays a crucial regulatory role inside the cell and that aberrant expression can precipitate cellular abnormalities [140].

## 6. Conclusions

Ran determines the cellular fate and regulates key activities for proper operation. First, Ran regulates the nucleocytoplasmic trafficking of RNAs, RNPs, proteins, and transcription factors, affecting gene transcription [53,141,142]. The overexpression of Ran in tumor cells may change nucleocytoplasmic transport and transcription factor distribution [6]. Tumor cell mitosis is abnormally reliant on the Ran pathway, and silencing it leads to dysregulated spindle formation, improper chromosomal segregation, and cell death [21,22,120]. This indicates a possible mechanism for the development of cancer abnormalities, therefore showing that Ran is vital for tumor cell survival. Additionally, tumor cells favor alternate Ran-dependent survival pathways [21,22]. Thus, Ran overexpression can lead to alterations in cellular proliferation, differentiation, and invasion, hallmarks of tumor cells and associated with malignancy. Ran is linked to malignancies [26,27]. Overexpression has been seen in breast cancer, ovarian cancer, melanoma, renal clear cell carcinoma, and lung cancer [21,22]. The overexpression of Ran may be dependent on upstream influences. This increases cellular dependency on Ran for survival, as shown by apoptosis upon Ran silencing [6].

High RAN has affected patient survival times. High Ran expression in patients with malignancies overexpressing osteopontin or with mutations that activate the PI3K and MEK signaling pathways is related to worse survival [6]. Overexpression has been correlated with increased malignancy and a worse prognosis in individuals with renal cell carcinoma. Moreover, Ran expression was correlated with prognosis and survival time in lung cancer and breast cancer patients. High expression is associated with a lower survival time and a longer disease-free survival time following surgery. From this, it follows that Ran regulation and signaling play a vital role in maintaining normal cell phenotypes, that its aberrant expression drives cellular disorders [143], and that it mediates cancer onset, tumor development, and severity [96].

## Figures and Tables

**Figure 1 ijms-24-03065-f001:**
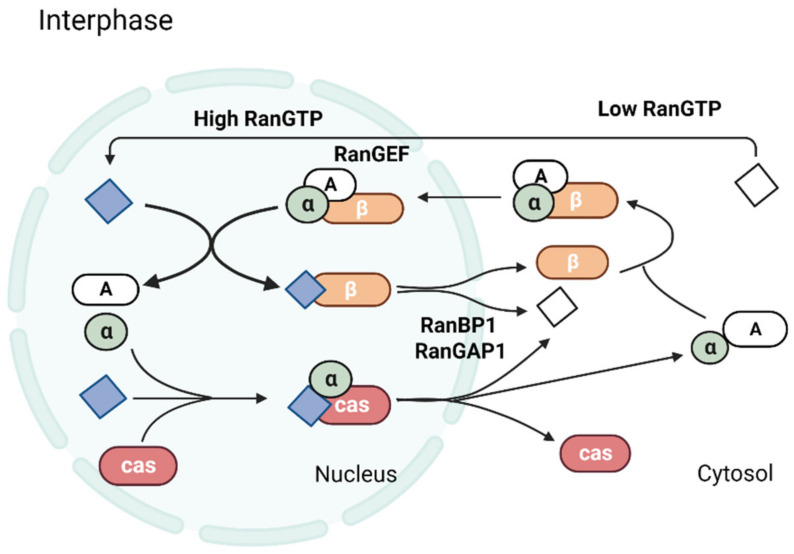
A complex containing aster-promoting activities, APA is the microtubule-associated protein NuMA (A), importin α (α), and importin β (β) forms in the cytosol and translocates across the nuclear pore. In the nucleus, Ran-GTP (filled diamonds, purple) binds to importin β and dissociates the transport complex. Ran-GTP and importin β shuttle back to the cytosol, where Ran-GTP is hydrolyzed by cytosolic RanGAP1 and RanBP1 to Ran-GDP (open diamonds, white). CAS (cas) and Ran-GTP bind to importin α in the nucleus, and this complex shuttles back to the cytosol, where it is also dissociated by the action of RanGAP1 and RanBP1. The polarized distribution of Ran-GTP across the nuclear envelope is maintained by the compartmentalization of RCC1 (indicated as RanGEF), RanGAP1, and RanBP.

**Figure 2 ijms-24-03065-f002:**
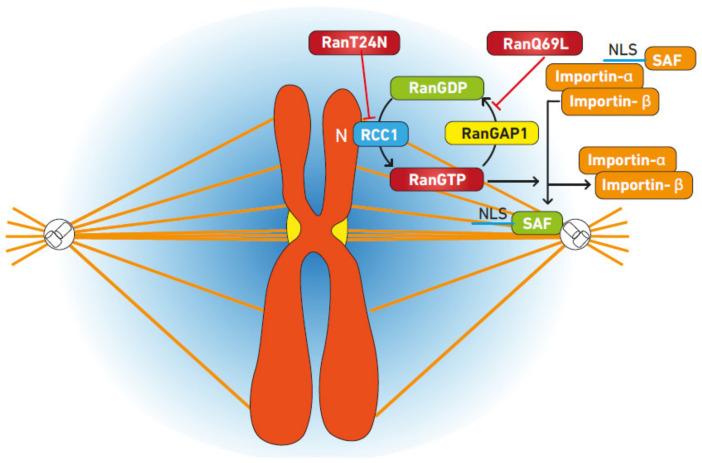
Signaling pathways of Ran GTP during mitotic spindle assembly. Generating RanGTP from RanGDP by the guanine nucleotide-exchange factor RCC1 on chromosomes produces a ‘cloud’ of RanGTP around the chromosomes. The gradient of this cloud is reduced distally to the chromosomes, where GTP is hydrolyzed by Ran, which has been stimulated by the Ran GTPase-activating protein RanGAP1. Experimentally, RanGTP production is inhibited by the mutant RanT24N, whereas another mutant, RanQ69L, is resistant to GTP hydrolysis and increases RanGTP concentrations distal to chromosomes. Around chromosomes, the cloud of RanGTP (illustrated by the shading) causes the release of spindle assembly factors (SAFs) from inhibitory complexes with importin-α and importin-β, which bind to a nuclear localization sequence (NLS) on a SAF and prevent its interaction with other proteins or otherwise inhibit SAF activity.

**Figure 3 ijms-24-03065-f003:**
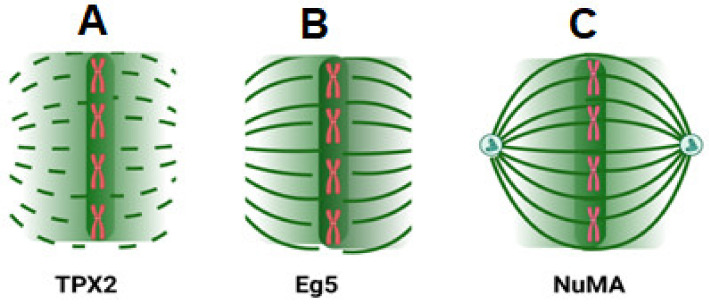
Mitotic proteins thought to be controlled by Ran GTP and their function in spindle assembly. (**A**) Nucleation: Chromosomes produce a Ran GTP-rich environment in the early stages. The importins release TPX2, which then initiates microtubule nucleation. Assembled microtubules surrounding the chromosomes begin to condense and arrange in (**B**). Eg5 and Kid, two proteins thought to be controlled by Ran GTP and involved in these processes, have been proposed. In spindle assembly, microtubules’ minus ends concentrate towards their poles, forming the third stage, called pole formation (**C**). Ran GTP regulates this process by bringing together NuMA and XCTK2. It is possible that TPX2 is involved as well.

## Data Availability

Not applicable.

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
