# Peer review of "Ran GTPase and Its Importance in Cellular Signaling and Malignant Phenotype"

_ijms, 2023, doi:10.3390/ijms24043065_

Round 1
Reviewer 1 Report
this review discusses an important subject on the role of Ran in cancer initiation and development when overexpressed in the cells.
The English language needs to be revised and give the full name for the abbreviations you mentioned like PI3K/Akt 234 and MEK/ERK Pathways.
Author Response
On behalf of the authors, I should like to thank the reviewers for taking the time and making the effort to give us insightful comments and recommendations. The reviewers' remarks have significantly improved the manuscript's quality. I hope that the work now meets the standards of the reviewers and is acceptable for publication in IJMS.
Response to Reviewer #1:
- We thank the respected reviewer for considering this work an important subject on the role of Ran in cancer initiation and development when overexpressed in the cells. Herein you can see our point-by-point responses according to the reviewers' comments:The English language needs to be revised and give the full name for the abbreviations you mentioned like PI3K/Akt 234 and MEK/ERK Pathways.
Author Response: The English language was completely rewritten and enhanced, and the full names of medical terminology were provided when they appeared for the first time in the text. [Page 1, lines 20-22] are marked in yellow, for example.
“Extracellular-regulated kinase/mitogen-activated protein kinase (ERK/MEK) and phosphatidylinositol 3-kinase / Protein kinase B (PI3K/Akt) pathways.”

Reviewer 2 Report
The manuscript by El-Tanani et al. reviews Ran GTPase in human cancer. This is an interesting topic given the importance of Ran in nuclear trafficking and how its activity is hijacked in human cancers. However, the manuscript in its current form is not suitable for publication as it needs serious editing to remove the many repetitions and the irrelevant details that are unnecessary in a review article especially if they don't have a clear message (see end of sentence on line 443 for one of many example). In its current form, the manuscript repeats the literature without synthesizing it and does not go beyond it.
A second major deficiency of the manuscript is that the authors start with the importance of Ran from cellular studies and end up with patient data. It is advisable to start with survival data to show importance of Ran in human cancer and then describe the function/contribution of Ran to cancer from cellular data.
Thirdly, the figures have no message and don’t contribute to the overall message. They need to be redone and reflect the message of the review.
Few suggestions that might help with the overall presentation:
- In the intro, the authors describe effects in cells and conclude that the patient’s data follows this trend. It should be the other way around.
- Explain abbreviations the first time you use them, e.g. NLS, NES and many others.
- Figure 1 is difficult to read. Please explain on the right or left of the figure what each symbol (colored and white squares, A, etc.) means.
- Under section 3, ‘Ran regulates spindle formation’ Figure 2 is supposed to show that ‘upon inhibition of the formation of RanGTP, the ability of chromatin to nucleate microtubules was completely removed’ (line 158). But Figure 2 doesn't show that. As shown this figure doesn’t bring anything new.
- Line 183: explain what SAF is.
- Line 209: explain what NE is.
- On line 138 the authors claim that RanT24N preferentially binds to GDP but on line 219 they say that the T24N reduces binding of both GDP and GTP. The authors should make up their mind. The T to N mutation interferes with Mg2+ binding and reduces the affinity of both di- and tri-nucleotide phosphates.
- The section lines 215-225 is too technical and not necessary in a review article as it derails the reader. Just summarize the message.
- Line 230: what are FxFG nucleoporins?
- Line 245: the authors should specify cancer cells if they mean cancer cells.
- Lines 259-260: change to ‘c-Met has been implicated in various cancers including lung, breast, and colon [121]’.
- Line 252: which KRAS mutant?
- Line 254: it is PI3K and not P13K.
- In section 4, adding a figure that illustrates the signaling pathways responsible for Ran gradient including all the mentioned proteins would be helpful.
- The sentence lines 310-315 is not clear. Is it missing ‘as’ before Ran on line 312?
- Line 371, what is EOC?
- Lines 376-383, details are not necessary in a review article. Summarize the findings.
- Line 449: the sentence starts with cell line HCC827 but finishes with survival of cancer patients with hhigh c-Met. Please edit and correct.
- It is preferred to start with section 5.2.2 indicating that high Ran expression correlates with worse prognosis than move to cellular work.
Author Response
On behalf of the authors, I should like to thank the reviewers for taking the time and making the effort to give us insightful comments and recommendations. The reviewers' remarks have significantly improved the manuscript's quality. I hope that the work now meets the standards of the reviewers and is acceptable for publication in IJMS.
Response to Reviewer #2:
Given the significance of Ran in nuclear trafficking and how its activity is hijacked in human tumors, we would like to thank the expert reviewer for considering our work to be an interesting issue. Here you will find our point-by-point answers to the reviewers' remarks:
- The manuscript in its current form is not suitable for publication as it needs serious editing to remove the many repetitions and the irrelevant details that are unnecessary in a review article especially if they don't have a clear message (see end of sentence on line 443 for one of many example). In its current form, the manuscript repeats the literature without synthesizing it and does not go beyond it.
Author Response: The manuscript has been entirely rewritten/rephrased and any needless repetitions have been deleted. [ Follow the tracked changes copy for insertion and deletion sections]. For example, Section 5.2.2. entitled “Ran expression and survival time” is now became section 2.2.2. and the sentence mentioned above in line 443 is now rephrased and clarified, lines 245-257, to be “This was also demonstrated to be independent of other parameters, including tumor size, grade, and lymph node involvement.”
- The authors start with the importance of Ran from cellular studies and end up with patient data. It is advisable to start with survival data to show importance of Ran in human cancer and then describe the function/contribution of Ran to cancer from cellular data.
Author Response: The manuscript has been completely re-written/ rephrased. Section 5 that discussing the “Effect of Ran Expression on Spindle Formation and tumor Cell Survival” which mentioned at the end of the manuscript is now became at the beginning in section 2 [page 3 line 136] directly after the introduction. Followed by other sections that describe the function/contribution of Ran to cancer from cellular data. [From page 6 line 256 to the end of the manuscript]
- The figures have no message and don’t contribute to the overall message. They need to be redone and reflect the message of the review.
Author Response: Figure 1 has now new description even in the text [ page 6 lines 262-267] or in legend [pages 7 and 8 lines 318-326]. Moreover, Figure 2 and figure 3 were also indicated in the text [page 9 line 383 and page 11 line 459, respectively] with a new Figure 2 and modified legend.
- In the intro, the authors describe effects in cells and conclude that the patient’s data follows this trend. It should be the other way around.
Author Response: We thank the reviewer for his/her insightful thoughts. We've corrected the introduction. It was rewritten as follows [page 2 lines 53-91].:
“Ran has therefore been implicated in multiple human cancers cell lines [9, 26-28]. It is thought to play a contributory role in tumor formation in various types of human cancer. Implantation of Ran overexpressing cells has been shown to be tumorigenic in vivo [29]. Furthermore, overexpression of Ran has been observed in numerous human cancer cell lines [9, 26-28]. Ran is an effector molecule of several growth factors which have been shown to play a significant part in some cancer types [30, 31]. Any changes in the expression of these genes have a downstream effect on Ran expression, therefore suggesting a pathway for tumourigenesis [9, 26, 32, 33].
These alterations may play an important role in cancer, resulting in the formation of tumors and a rise in malignancy in correlation to Ran's overexpression [34]. The aberrant expression of Ran causes the dysregulation of several processes within the cells, leading to altered cellular function. Cells expressing high Ran levels display several fundamental changes; an increased ability to proliferate, including the loss of contact inhibition, which leads to increased tissue density and the ability to grow in an anchorage-independent manner [3, 26, 28, 30, 35, 36].”
- Explain abbreviations the first time you use them, e.g. NLS, NES and many others.
Author Response: All abbreviations were described and clarified at first mentioned.
For example, NLSs stands for nuclear localization signals and first mentioned at [page 2 line 70].NES stands for nuclear export signal and first mentioned at [page 2 line 70]. RCC1 stands for regulator of chromosome condensation 1 and first mentioned at [page 2 line 56]. TPX2 stands for targeting protein X2 and first mentioned at [page 3 line 142]. siRNA stands for small interfering RNA and first mentioned at [page 3 line 147].OPN stands for oeteopontin and first mentioned at [page 1 line 20]. c-Met stands for mesenchymal epithelial transition factor and first mentioned at [page 5 line 247].
- Figure 1 is difficult to read. Please explain on the right or left of the figure what each symbol (colored and white squares, A, etc.) means.
Author Response: Figure 1 has now new description even in the text [ page 6 lines 262-268] or in legend [pages 7 and 8 lines 318-326]. As follow:
[“Figure 1. During interphase, Ran enhances nuclear localization signals (NLS)-mediated protein import of aster promoting activities (APA) components by forming and dissociation transport complexes. A complex composed of aster-promoting activities, importin α (α), and importin β (β) formed in the cytosol and translocates through the nuclear pore. Ran-GTP (filled diamonds) binds to importin and dissociates the transport complex in the nucleus. Ran-GTP and importin return to the cytosol, where Ran-GTP is converted to Ran-GDP by cytosolic RanGAP1 and RanBP1 (open diamonds). importin β–related export receptor (cas) and Ran-GTP bind to importin α in the nucleus, and this complex shuttles back to the cytosol, where RanGAP1 and RanBP1 dissociate it. The compartmentalization of RCC1 (shown as RanGEF), RanGAP1, and RanBP preserves the polarized distribution of Ran-GTP across the nuclear envelope.]
- Under section 3, ‘Ran regulates spindle formation’ Figure 2 is supposed to show that ‘upon inhibition of the formation of RanGTP, the ability of chromatin to nucleate microtubules was completely removed’ (line 158). But Figure 2 doesn't show that. As shown this figure doesn’t bring anything new.
Author Response: Apologies for any confusion caused by the incorrect citation in the text, which has been rectified as follows [ page 9 lines 381-383].:
It is assumed that chromatin plays a role in spindle formation [105, 107]. Ran GTP has a concentration gradient surrounding chromosome, which is mediated by RCC1's association with chromatin (modified Figure 2).
- Line 183: explain what SAF is.
Author Response: Done, SAFs stand for spindle assembly factors and its first mentioned now in [ page 10 line 429].
- Line 209: explain what NE is.
Author Response: Done, NE stand for nuclear envelope and its first mentioned in the abstract in [ page 1 line 18].
- On line 138 the authors claim that RanT24N preferentially binds to GDP but on line 219 they say that the T24N reduces binding of both GDP and GTP. The authors should make up their mind. The T to N mutation interferes with Mg2+ binding and reduces the affinity of both di- and tri-nucleotide phosphates.
Author Response: The information has been appropriately restated. [ page 9 line 369]. As follow:
“Ran GTP has been demonstrated to induce the production of microtubule asters surrounding centrosomes in vitro, whereas RanT24N, which is predominantly coupled to GDP, lacks this capacity”
- The section lines 215-225 is too technical and not necessary in a review article as it derails the reader. Just summarize the message.
Author Response: The lengthy details have been condensed into one paragraph. [ page 12 lines 507-512]. As follow:
“Mitotic division also requires regulated reassembly of the nuclear membrane, which separates the nucleus and cytoplasm. Ran GTPase regulates this process. NE construction requires both RCC1-generated RanGTP and Ran's GTPase activity [25, 131]. Ran depletion in C. elegans induced NE defects. This data implicates the Ran cycle as a modulator of NE reassembly, either by vesicle fusion or chromatin modification [101]. Ran induces NPC-containing NE assembly in sepherose beads without chromatin [132].”
- Line 230: what are FxFG nucleoporins?
Author Response: Done, FxFG is stands for Phe-any-Phe-Gly nucleoporins and it’s deleted during re-writing process.
- Line 245: the authors should specify cancer cells if they mean cancer cells.
Author Response: The paragraph was amended per reviewer request [ page 13 lines 555-559]. As follow:
“When breast cancer cells were treated with separate inhibitors for both the PI3K and MEK pathways, silencing of Ran resulted in a greater degree of apoptosis than under normal conditions, suggesting that the abnormal activation of these pathways leads to tumour cell dependence on Ran [6, 143, 144].”
- Lines 259-260: change to ‘c-Met has been implicated in various cancers including lung, breast, and colon [121]’.
Author Response: The sentence was corrected, [ page 14 lines 581]. As follow:
“Moreover, c-Met has been linked to lung, breast, colon cancer, and others [149].”
- Line 252: which KRAS mutant?
Author Response: This sentence was omitted from the section containing this text.
- Line 254: it is PI3K and not P13K.
Author Response: Done, its corrected to PI3K [ page 13 lines 579].
- In section 4, adding a figure that illustrates the signaling pathways responsible for Ran gradient including all the mentioned proteins would be helpful.
Author Response: A new graphic (Figure 2) describing the signaling pathways of Ran GTP during mitotic spindle construction has been created. [ page 10 lines 421-431].
- The sentence lines 310-315 is not clear. Is it missing ‘as’ before Ran on line 312?
Author Response: The paragraph was corrected, [ page 15 lines 672-674]. As follow:
“This dysregulation appears to be dependent on active PI3K/Akt and MEK/ERK pathways in cancer cells. Ran silencing increased nuclear localization of p53, p27, and C-jun while decreasing nuclear localization of β-catenin. In breast cancer cells, Ran silencing decreased -catenin and NFκB nuclear localization while increasing p53 and p27 localization[165, 166].”
- Line 371, what is EOC?
Author Response: EOC stand for epithelial ovarian cancer and its first mentioned now in [ page 4 line 154].
- Lines 376-383, details are not necessary in a review article. Summarize the findings.
Author Response: The findings were rewritten and summarized [ page 4 line 180-186].
“Overexpression of Ran was elevated at both the mRNA and protein levels in gastric adenocarcinomas. In contrast, normal cells within the Tumor tissue did not exhibit higher levels of Ran [49]. In mouse tissue, it was discovered that lymphoma cells entering the liver had elevated levels of Ran, whereas surrounding hepatocytes had normal levels. In addition, Ran expression was found to be significantly high in a variety of human malignancies compared to normal tissue, such as in colon, breast, lung, and renal cell adenocarcinomas [22, 23].”
- Line 449: the sentence starts with cell line HCC827 but finishes with survival of cancer patients with high c-Met. Please edit and correct.
Author Response: The sentence was edited and corrected [ page 5 line 245-247] as follow:
“In lung cancer, increased Ran expression drastically decreased the median survival time of individuals who overexpress the mesenchymal epithelial transition factor (c-Met) [62, 63]. “
- It is preferred to start with section 5.2.2 indicating that high Ran expression correlates with worse prognosis than move to cellular work.
Author Response: We replaced the intended section as follow:
Section 2: Effect of Ran Expression on Spindle Formation and tumor Cell Survival [ page 3 line 136] Followed by other sections that describe the function/contribution of Ran in human cancer. [From page 6 line 259 to the end of the manuscript]

Reviewer 3 Report
Tanani et al. review the importance of Ran, a small Ras GTPase, in the cell cycle and in malignant tumour cells.
In general the topic is of interest, but the way the authors present data which have been published in literature, is not comprehensive and convincing. The reader is completely confused by the flood of data which are just annotated without any critical selection. A lot of abbreviations remain unexplained like NuMA, STAT, TPX2.
Chapter 2 and chapter 3 are more than complicated to read.The reader is lost by all the detailed informations. These informations have to be simplified and structured according to their importance to reach a broad readership. A summary of experimental data is presented while necessary interpretations of the data is absent
The depicted figures in particular figures 1 and 2 are not self explaining. Please extend the legend of Fig.1 since the reader cannot follow what is described in the text. Moreover, chapters 2 and 3 have to be simplified and rewritten.
Fig.2 does not tell what is written in the legend. I am not sure at all what this figure reflects. GEF remains unexplained and RanBP1 does not appear in the Figure. This is also the case for Fig.3.
The Conclusion section repeats passages of the text and is far too long. Instead it should be very concise and critical.A perspective should be given.
Minor points:
A lot of difficult sentences appear in this manuscript which cannot be understood. I would suggest to involve a native speaker to solve these problems in grammar and spellings.
Lane 144: “Upon the addition of Ran mutants” This does not sound scientifically-what is the meaning?
Lane 179: “As we have already talked about” Such a term should not appear in a scientific manuscript.
Lanes 212-214: “In conjunction with the fact that RanG- 212 TPs localized concentration around mitotic chromatin, this evidence suggests that Ran 213 may function to recruit vesicles to chromatin and initiate their fusion “ What is the meaning of this sentence?
Lanes 472-473: ….suggests a potential mechanism for the production of the abnormalities 472 displayed in cancer. This evidence suggests that Ran is essential for tumour cell survival. “ A suggestion can never be an evidence
The manuscript contains a lot of repeats and not precise informations.These are only some examples.
My recommendation is a complete and major revision of this manuscript.
Author Response
On behalf of the authors, I should like to thank the reviewers for taking the time and making the effort to give us insightful comments and recommendations. The reviewers' remarks have significantly improved the manuscript's quality. I hope that the work now meets the standards of the reviewers and is acceptable for publication in IJMS.
Response to Reviewer #3:
We would like to thank the respected reviewer for considering our manuscript of interest. Herein you can see our point-by-point responses according to the reviewers' comments:
- The reader is completely confused by the flood of data which are just annotated without any critical selection.
Author Response: The manuscript has been completely rewritten/ corrected along with removing any unnecessary repetitions [ Follow the tracked changes copy for insertion and deletion sections]. For example, Section 5.2.2. entitled “Ran expression and survival time” is now became section 2.2.2. Section 5 that discussing the “Effect of Ran Expression on Spindle Formation and Tumour Cell Survival” which mentioned at the end of the manuscript is now became at the beginning in section 2 [ page 3 line 136] directly after the introduction. Followed by other sections that describe the function/contribution of Ran to cancer from cellular data. [From page 6 line 259 to the end of the manuscript]
- A lot of abbreviations remain unexplained like NuMA, STAT, TPX2.
Author Response: All abbreviations were described and clarified when first mentioned.
For example, NLSs stands for nuclear localization signals and first mentioned at [page 2 line 69].NES stands for nuclear export signal and first mentioned at [page 2 line 70]. RCC1 stands for regulator of chromosome condensation 1 and first mentioned at [page 2 line 56]. TPX2 stands for targeting protein X2 and first mentioned at [page 3 line 142]. siRNA stands for small interfering RNA and first mentioned at [page 3 line 147].OPN stands for oeteopontin and first mentioned at [page 1 line 20]. c-Met stands for mesenchymal epithelial transition factor and first mentioned at [page 5 line 250]. NE stand for nuclear envelope and its first mentioned in the abstract in [ page 1 line 17]. SAFs stand for spindle assembly factors and its first mentioned now in [ page 10 line 450]. NuMA stands for nuclear mitotic apparatus protein its first mentioned now in [ page 10 line 436].STAT-3 stands for signal transducer and activator of transcription 3 its first mentioned now in [ page 8 line 344].
- Chapter 2 and chapter 3 are more than complicated to read. The reader is lost by all the detailed informations. These informations have to be simplified and structured according to their importance to reach a broad readership. A summary of experimental data is presented while necessary interpretations of the data is absent.
Author Response: The manuscript has been completely rewritten/ corrected along with removing any unnecessary repetitions and complications [ Follow the tracked changes copy for insertion and deletion sections]. For example, Section 5.2.2. entitled “Ran expression and survival time” is now became section 2.2.2. Section 5 that discussing the “Effect of Ran Expression on Spindle Formation and Tumour Cell Survival” which mentioned at the end of the manuscript is now became at the beginning in section 2 [ page 3 line 136] directly after the introduction. Followed by other sections that describe the function/contribution of Ran to cancer from cellular data. [From page 6 line 259 to the end of the manuscript]. Moreover, introduction section was completely summarized in which last two paragraphs were removed [ page 3 line 121-134]. Furthermore, section 3.1 (Molecules Transported) was re-written, and the end paragraph was eliminated [ page 8 line 365-369].
- The depicted figures in particular figures 1 and 2 are not self-explaining. Please extend the legend of Fig.1 since the reader cannot follow what is described in the text.
Author Response: Figure 1 has now new description even in the text [ page 6 lines 266-271] or in legend [pages 7 and 8 lines 322-330]. As follow:
[“Figure 1. During interphase, Ran enhances nuclear localization signals (NLS)-mediated protein import of aster promoting activities (APA) components by forming and dissociation transport complexes. A complex composed of aster-promoting activities, importin α (α), and importin β (β) formed in the cytosol and translocates through the nuclear pore. Ran-GTP (filled diamonds) binds to importin and dissociates the transport complex in the nucleus. Ran-GTP and importin return to the cytosol, where Ran-GTP is converted to Ran-GDP by cytosolic RanGAP1 and RanBP1 (open diamonds). importin β–related export receptor (cas) and Ran-GTP bind to importin α in the nucleus, and this complex shuttles back to the cytosol, where RanGAP1 and RanBP1 dissociate it. The compartmentalization of RCC1 (shown as RanGEF), RanGAP1, and RanBP preserves the polarized distribution of Ran-GTP across the nuclear envelope.] Moreover, new figure 2 and figure 3 were also indicated in the text [page 9 line 387 and page 10 line 445, respectively] and were fully explained in the legands.
- Chapters 2 and 3 have to be simplified and rewritten.
Author Response: The manuscript has been completely rewritten/ rephrased along with removing any unnecessary repetitions and complications [ Follow the tracked changes copy for insertion and deletion sections]. For example, Section 5.2.2. entitled “Ran expression and survival time” is now became section 2.2.2. Section 5 that discussing the “Effect of Ran Expression on Spindle Formation and Tumour Cell Survival” which mentioned at the end of the manuscript is now became at the beginning in section 2 [ page 3 line 136] directly after the introduction. Followed by other sections that describe the function/contribution of Ran to cancer from cellular data. [From page 6 line 259 to the end of the manuscript]. Moreover, introduction section was completely summarized in which last two paragraphs were removed [ page 3 line 121-134]. Furthermore, section 3.1 (Molecules Transported) was re-written, and the end paragraph was eliminated [ page 8 line 365-369].
- 2 does not tell what is written in the legend. I am not sure at all what this figure reflects. GEF remains unexplained and RanBP1 does not appear in the Figure. This is also the case for Fig.3.
Author Response: A new figure has been created to replace the previous one, and the legend has been revised to describe the signaling routes of Ran GTP during mitotic spindle formation.
- The Conclusion section repeats passages of the text and is far too long. Instead, it should be very concise and critical. A perspective should be given.
Author Response: Done, the conclusion was edited and corrected [ page 48 line 863-851] as follow:
Ran determines cellular fate and regulates key activities for proper operation. First, Ran regulates the nucleocytoplasmic trafficking of RNAs, RNPs, proteins, and transcription factors, affecting gene transcription [88, 171, 172]. Overexpression of Ran in tumor cells may change nucleocytoplasmic transport and transcription factor distribution. This may result in pregnancy irregularities [6]. Tumour cell mitosis is abnormally reliant on the Ran pathway, and silencing it leads to dysregulated spindle formation, improper chromosomal segregation, and cell death [22, 23, 41]. This indicates a possible mechanism for the development of cancer abnormalities. This shows Ran is vital for tumor cell survival. Additionally, tumor cells favor alternate Ran-dependent survival pathways [22, 23]. Therefore, Ran overexpression can lead to alterations in cellular proliferation, differentiation, and invasion, hallmarks of tumor cells and associated with malignancy. Ran is linked to malignancies [28, 29]. Overexpression has been seen in breast cancer, ovarian cancer, melanoma, renal clear cell carcinoma, and lung cancer [22, 23]. The overexpression of Ran may be dependent on upstream influences. This increases cellular dependency on Ran for survival, as shown by apoptosis upon Ran silencing [6].
High RAN has affected patient survival times. High Ran expression in patients with malignancies overexpressing osteopontin or with mutations that activate the PI3K and MEK signaling pathways related to worse survival [6, 165, 173] . Overexpression has been correlated with increased malignancy and a worse prognosis in individuals with renal cell carcinoma. Moreover, Ran expression was correlated with prognosis and survival time in lung cancer and breast cancer patients. High expression is associated with a lower survival time and a longer disease-free survival time following surgery. From this, it follows that Ran regulation and signaling play a vital role in maintaining normal cell phenotypes and that its aberrant expression drives cellular disorders [174] and that it mediates cancer onset, tumor development, and severity [37].
Minor points:
- A lot of difficult sentences appear in this manuscript which cannot be understood. I would suggest involving a native speaker to solve these problems in grammar and spellings.
Author Response: A native tongue speaker completely altered and enhanced the English language. The manuscript has been entirely rewritten/corrected and any needless repetitions and complexity have been eliminated. [ Follow the tracked changes copy for insertion and deletion sections].
- Lane 144: “Upon the addition of Ran mutants” This does not sound scientifically-what is the meaning?
Author Response: The sentence was edited and corrected [ page 8 line 375-376] as follow: “Little centrosomal aster development was detected after the inclusion of the Ran mutant [103].”
- Lane 179: “As we have already talked about” Such a term should not appear in a scientific manuscript.
Author Response: The sentence was edited and corrected [ page 1 line 446-448] as follow: “As previously discussed, this results in the formation of microtubules surrounding chromosomes [122-125]”
- Lanes 212-214: “In conjunction with the fact that RanG- 212 TPs localized concentration around mitotic chromatin, this evidence suggests that Ran 213 may function to recruit vesicles to chromatin and initiate their fusion “ What is the meaning of this sentence?
Author Response: Done, the section 4.2. (Ran Regulates Nuclear Envelope Reassembly) was edited and corrected [ page 12 line 507-512] and this paragraph was deleted. the section was summarized and clarified to the following paragraph “Mitotic division also requires regulated reassembly of the nuclear membrane, which separates the nucleus and cytoplasm. Ran GTPase regulates this process. NE construction requires both RCC1-generated RanGTP and Ran's GTPase activity [25, 131]. Ran depletion in C. elegans induced NE defects. This data implicates the Ran cycle as a modulator of NE reassembly, either by vesicle fusion or chromatin modification [101]. Ran induces NPC-containing NE assembly in sepherose beads without chromatin [132].“
- Lanes 472-473: ….suggests a potential mechanism for the production of the abnormalities 472 displayed in cancer. This evidence suggests that Ran is essential for tumour cell survival. “A suggestion can never be an evidence.
Author Response: The sentence was edited and corrected [ page 19 line 847-848] as follow: “This indicates a possible mechanism for the development of cancer abnormalities. This shows Ran is vital for tumor cell survival.”
- The manuscript contains a lot of repeats and not precise informations. These are only some examples.
Author Response: The manuscript has been completely rewritten/ rephrased along with removing any unnecessary repetitions and complications [ Follow the tracked changes copy for insertion and deletion sections]. For example, Section 5.2.2. entitled “Ran expression and survival time” is now became section 2.2.2. Section 5 that discussing the “Effect of Ran Expression on Spindle Formation and Tumour Cell Survival” which mentioned at the end of the manuscript is now became at the beginning in section 2 [ page 3 line 136] directly after the introduction. Followed by other sections that describe the function/contribution of Ran to cancer from cellular data. [From page 6 line 259 to the end of the manuscript]. Moreover, introduction section was completely summarized in which last two paragraphs were removed [ page 3 line 121-134]. Furthermore, section 3.1 (Molecules Transported) was re-written, and the end paragraph was eliminated [ page 8 line 365-369].

Round 2
Reviewer 3 Report
The manuscript has improved in parts after revision. Hence, the information provided is not structured and organized. This is a major concern that appears in the whole manuscript. It seems that the information is just annotated without the main focus. In this context lines 67 to 73 are not necessary as an example. The reader gets lost and makes the manuscript very hard to read. A lot of abbreviations still remain unexplained. I strongly recommend an appendix or a list of abbreviations.
Figures 1 and 2 are not self explaining- changes in the figures have to be performed. Fig.1: What does alpha, beta mean? What do the purple diamonds stand for?
Fig.2: Where is A, B, C as described in the legend?
What does NUMA stand for? These informations should be given in the legend.
The manuscript can not be published after the first revision. I strongly recommend to get support by a native speaker since there are many grammatical errors and spelling mistakes.
Line 51 "impact upon"
Line 144 spelling of tumours
line 362 "by " instead of in the mutant
Author Response
Notes to the respected editor:
On behalf of the authors, I should like to thank the reviewers for taking the time and making the effort to give us insightful comments and recommendations. The reviewers' remarks have significantly improved the manuscript's quality. I hope that the work now meets the standards of the reviewers and is acceptable for publication in IJMS.
Response to Reviewer # 3 (round 2):
We would like to thank the respected reviewer for considering our manuscript improved after first revision. Herein you can see our point-by-point responses according to the reviewers' comments:
1- The information provided is not structured and organized. This is a major concern that appears in the whole manuscript. It seems that the information is just annotated without the main focus. In this context lines 67 to 73 are not necessary as an example. The reader gets lost and makes the manuscript very hard to read.
Author Response: We respect the reviewer major comment, we apologize that the information seems annotated without focus and makes the manuscript very hard to read.
The whole manuscript has been completely restructured / re-organized in the following ordered outline (All movements were highlighted in pink in the attached manuscript):
- Introduction:
1.2The roles of Ran within the cell
1.2 The role of Ran in nucleocytoplasmic transport and cell cycle progression
1.3 The role of Ran in Cancer progression
- Ran Regulates Nucleocytoplasmic Transport
2.1 Molecules Transported
- Ran Regulates Spindle formation
3.1. Mechanism of Spindle Regulation by Ran
3.2. Ran Regulates Nuclear Envelope Reassembly
- The Overexpression of Ran Alters Cellular Growth and Proliferation and is Present in Cancer
4.1. Ran Overexpression
4.1.1. Ran Overexpression and Malignancy in Human Cancers
4.1.2. Ran expression and survival time
- Mechanism of Altered Expression– Ran is a Downstream Effector of the PI3K/Akt and MEK/ERK Pathways.
5.1. Aberrant Control of Pathways and Tumor Cell Dependence on Ran
5.2. Signaling Pathways - Phosphorylation of Ran binding proteins and Control of Ran Expression
5.3. Effect of Ran Expression on Spindle Formation and Tumor Cell Survival
- Conclusion
Our goal in structuring the article was to give the reader a sense of RanGTP's physiological function in normal cells, its role in cancer progression, and its mechanism of action in malignant transformation. We hope that the new format and structure is more clear, understandable and goes smoothly.
2- A lot of abbreviations still remain unexplained. I strongly recommend an appendix or a list of abbreviations.
Author Response: All abbreviations were described and clarified when first mentioned. We think that adding an appendix will cause repetition and lengthen the manuscript.
For example, NLSs stands for nuclear localization signals and first mentioned at [page 2 line 54].NES stands for nuclear export signal and first mentioned at [page 2 line 54]. RCC1 stands for regulator of chromosome condensation 1 and first mentioned at [page 1 line 41]. TPX2 stands for targeting protein X2 and first mentioned at [page 4 line 459]. siRNA stands for small interfering RNA and first mentioned at [page 6 line 251]. OPN stands for osteopontin and first mentioned at [page 1 line 21]. c-Met stands for mesenchymal epithelial transition factor and first mentioned at [page 7 line 299]. NE stand for nuclear envelope and its first mentioned in the abstract in [ page 1 line 18]. SAFs stand for spindle assembly factors and its first mentioned now in [ page 5 line 170]. NuMA stands for nuclear mitotic apparatus protein its first mentioned now in [ page 6 line 179].STAT-3 stands for signal transducer and activator of transcription 3 its first mentioned now in [ page 4 line 131]. All page numbering and lines in the clean copy.
3- Figures 1 and 2 are not self explaining- changes in the figures have to be performed. Fig.1: What does alpha, beta mean? What do the purple diamonds stand for? Fig.2: Where is A, B, C as described in the legend?
Author Response: Figure 1 has now new description even in the text [ page 2 lines 65] or in legend [pages 3 lines 110-118]. Moreover, Figure 2 and figure 3 were also indicated in the text [page 4 line 153 and page 5 line 200, respectively]. Moreover, Figure 1 and figure 2 were have an new and modified legends describing each elements mentioned in the figures as follow. [pages 3 lines 110-118] and [pages 4 and 5 lines 163 -172] in the clean copy.
Figure 1. A complex containing aster promoting activities, importin α (α) and importin β (β) forms in the cytosol and translocates across the nuclear pore. In the nucleus, Ran-GTP (filled diamonds, purple) binds to importin β and dissociates the transport complex. Ran-GTP and importin β shuttle back to the cytosol, where Ran-GTP is hydrolyzed by cytosolic RanGAP1 and RanBP1 to Ran-GDP (open diamonds, white). CAS (cas) and Ran-GTP bind to importin α in the nucleus, and this complex shuttles back to the cytosol, where it is also dissociated by the action of RanGAP1 and RanBP1. The polarized distribution of Ran-GTP across the nuclear envelope is maintained by the compartmentalization of RCC1 (indicated as RanGEF), RanGAP1, and RanBP.
Figure 2. Signaling pathways of Ran GTP during mitotic spindle assembly.Generating RanGTP from RanGDP by the guanine nucleotide-exchange factor RCC1 on chromosomes produces a ‘cloud’ of RanGTP around the chromosomes. The gradient of this cloud is reduced distal to the chromosomes, where GTP is hydrolysed by Ran, which has been stimulated by the Ran GTPase-activating protein RanGAP1. Experimentally, RanGTP production is inhibited by the mutant RanT24N, whereas another mutant, RanQ69L, is resistant to GTP hydrolysis and raises RanGTP concentrations distal to chromosomes. Around chromosomes, the cloud of RanGTP (illustrated by the shading) causes the release of spindle assembly factors (SAFs) from inhibitory complexes with importin-α and importin-β, which bind to a nuclear localization sequence (NLS) on a SAF and prevent its interaction with other proteins or otherwise inhibit SAF activity.
4- What does NUMA stand for? These information should be given in the legend.
Author Response: All abbreviations were described and clarified when first mentioned.
NuMA stands for nuclear mitotic apparatus protein, and it is now mentioned in the clean copy at [ page 6 line 179].
5- The manuscript cannot be published after the first revision. I strongly recommend to get support by a native speaker since there are many grammatical errors and spelling mistakes. Line 51 "impact upon", Line 144 spelling of tumours, line 362 "by " instead of in the mutant
Author Response: The English language was completely altered and improved by the addition of a second native speaker. The manuscript has been completely rewritten/corrected, and any unnecessary repetitions and complexities have been removed. [Follow the tracked changes copy for section insertion and deletion]. Furthermore, "impact upon" was changed to "impact on" in Line 51. The spelling of tumours on line 144 has been corrected. Line 362 "by" instead of in the mutant was corrected, as shown in the new legend in figure 2.

Round 3
Reviewer 3 Report
Dear authors,
before this manuscript can be published a lot of spelling mistakes have to be corrected. This should not happen after a third revision!!
I am listing only a few:
line 142: phenotype transformation
there is an s occurring after tumors in lines: 142, 303, 307, 313, 290, 291??
Double dots reappear e.g. line 348.
Line 434: Remove this sentence about pregnancy. It hardly fits in the context of malignant cells.
Author Response
IJMS Journal
January 10, 2023
Subject: Response to reviewers
Dear Editor
In response to reviewers and editorial comments regarding our manuscript entitled "Ran GTPase and its Importance in Cellular Signaling and Malignant Phenotype " (Manuscript ID: ijms-2008567) which we submitted for publication in IJMS Journal, we carefully considered the comments offered by the respected reviewer 3. We thank the respected editor for considering our work.
Please see our point-by-point responses to reviewers' comments.
Sincerely,
Prof M El-Tanani
Notes to the respected editor:
On behalf of the authors, I should like to thank the reviewers for taking the time and making the effort to give us insightful comments and recommendations. I hope that the work now meets the standards of the reviewers and is acceptable for publication in IJMS.
Response to Reviewer # 3 (Round 3):
We would like to thank the respected reviewer for considering our manuscript improved after first revision. Herein you can see our point-by-point responses according to the reviewers' comments:
Before this manuscript can be published a lot of spelling mistakes have to be corrected. This should not happen after a third revision!! I am listing only a few:
Author Response: The English language was fully checked all over the manuscript, please follow tracked changes copy for section modification and deletion.
- Page 1 lines 39.
- Page 2 lines 50, 58, 68, 73, 80 and 87
- Page 3 lines 97,103,104,106 and 122.
- Page 4 lines 132,146, 148, 155, 160 and 161.
- Page 5 lines 177, 179, 182 and 191.
- Page 6 lines 205-2012, 214, 217-219, 228 and 232, 239.
- Page 7 lines 254, 271. 283, 187,288, 290 and 305.
- Page 8 lines 306, 314, 317, 320, 323 and 335, 325-346.
- Page 8 and 9 lines 349 -364.
- Page 9 lines 386, 391.
- Page 10 lines 429.
- line 142: phenotype transformation.
Author Response: Corrected to phenotype instead of phentype, it’s now in page 4 line 133.
- There is an s occurring after tumors in lines: 142, 303, 307, 313, 290, 291??
Author Response: Corrected whenever mentioned all over the text, lines 70, 75, 277, 278, 290, 294, 300, 389, 397, 399, 401, 404, and 426.
- Double dots reappear e.g., line 348.
Author Response: Corrected, it’s now in page 8 line 335.
- Line 434: Remove this sentence about pregnancy. It hardly fits in the context of malignant cells.
Author Response: The sentence related pregnancy irregularities was removed from page 10 line 421.
